# Sublethal concentrations of undissociated acetic acid may not always stimulate acid resistance in *Salmonella enterica* sub. *enterica* serovar Enteritidis Phage Type 4: Implications of challenge substrate associated factors

**Alkmini Gavriil** ⓘ**, Athina Thanasoulia, Panagiotis N. Skandamis***

Laboratory of Food Quality Control and Hygiene, Department of Food Science and Human Nutrition, Agricultural University of Athens, Athens, Greece

* pskan@aua.gr

## Abstract

Acid adaptation enhances survival of foodborne pathogens under lethal acid conditions that prevail in several food-related ecosystems. In the present study, the role of undissociated acetic acid in inducing acid resistance of *Salmonella* Enteritidis Phage Type 4 both in laboratory media and in an acid food matrix was investigated. Several combinations of acetic acid (0, 15, 25, 35 and 45 mM) and pH values (4.0, 4.5, 5.0, 5.5, 6.0) were screened for their ability to activate acid resistance mechanisms of pathogen exposed to pH 2.5 (screening assay). Increased survival was observed when increasing undissociated acetic acid within a range of sublethal concentrations (1.9–5.4 mM), but only at pH 5.5 and 6.0. No effect was observed at lower pH values, regardless of the undissociated acetic acid levels. Three combinations (15mM/pH5.0, 35mM/pH5.5, 45mM/pH6.0) were selected and further used for adaptation prior to inoculation in commercial tarama (fish roe) salad, i.e., an acid spread (pH 4.35 ± 0.02), stored at 5˚C. Surprisingly and contrary to the results of the screening assay, none of the acid adaptation treatments enhanced survival of *Salmonella* Enteritidis in the food matrix, as compared to non-adapted cells (control). Further examination of the food pH value, acidulant and storage (challenge) temperature on the responses of the pathogen adapted to 15mM/pH5.0, 35mM/pH5.5 and 45mM/pH6.0 was performed in culture media. Cells adapted to 35mM/pH5.5 were unable to induce acid resistance when exposed to pH 4.35 (tarama salad pH value) at 37˚C and 5˚C, whereas incubation under refrigeration (5˚C) at pH 4.35 sensitized 45mM/pH6.0 adapted cells against the subsequent acid and cold stress. In conclusion, pre-exposure to undissociated acetic acid affected the adaptive responses of *Salmonella* Enteritidis Phage Type 4 in a concentration- and pH-dependent manner, with regard to conditions prevailing during acid challenge.

**Data Availability Statement:** All relevant data are contained within the paper and its Supporting Information files.

**Funding:** The authors received no specific funding for this work.

**Competing interests:** The authors have declared that no competing interests exist.

## Introduction

It is now well documented that pre-exposure of microorganisms to sublethal stress conditions may induce adaptive responses that enhance resistance to subsequent lethal factors [1] of the same or multiple stresses (cross protection; [2]). Exposure to acid environments is frequently encountered by microorganisms during their route from the food chain to the human host. Among the foodborne pathogens, *Salmonella* spp., is able to induce resistance mechanisms as a result of adaptation to mild or moderate acid stress [3,4]. The acid induced phenotypic responses are highly affected by factors such as the selected strain [5,6] and conditions prevailing during adaptive and subsequent lethal challenge treatments, e.g., acidulant, temperature and composition [3]. So far, different adaptation protocols are used in order to formulate suitable sublethal acid conditions. For instance, acid adaptation can be achieved by supplementation of growth media with glucose [6,7] or by long- or short-term pre-exposure of cells to various organic or inorganic acidulants [8–13]. Nevertheless, it has been demonstrated that different protocols used to stimulate acid resistance can diversify the survival of *Escherichia coli* in apple juice stored under refrigeration [8].

Organic acids such as acetic can form acid stress conditions frequently encountered by *Salmonella* spp., as they are widely applied to the food industry. They are common preservatives in foods, such as mayonnaise and salad dressings, carcass decontamination treatment agents, whereas they can also accumulate in fermentable products as the result of indigenous or starter cultures activity. Finally, they are naturally present inside the gastrointestinal human or animal track due to the metabolic activity of endogenous microflora [11,14–18]. The antimicrobial activity of organic acids has been traditionally attributed to their undissociated molecules [17]. Despite the antibacterial efficiency of organic acids, their application might also pose considerable risk associated with potential induction of acid resistance. Thus, apart from adaptation to acids intrinsically encountered in foods, some food industry interventions may also promote induction of acid resistance [19], e.g., in environments with sublethal acid levels due to dilution of acid concentration with water, etc.. Stimulation of acid resistance mechanisms may result in increased likelihood of disease. A positive correlation between acid resistance and pathogenicity [5] has been found, alongside with evidence that acid adaptation increases virulence [20]. In addition, several regulators involved in the induced acid resistance of *Salmonella* spp. also control the expression of genes required for virulence [21].

So far, numerous studies have dealt with the responses of foodborne pathogens following adaptation to organic acids under different experimental conditions. Nevertheless, investigation pertaining the role of undissociated acid to the induction of acid resistance is limited. In addition, considering the protective effect of some food matrices on the ability of bacterial cells to tolerate lethal stresses [12,22], it is important to compare the results from experiments in laboratory media to those from food related environments.

Given the above, this study was conducted in order to examine the contribution of undissociated acetic acid over a range of different pH values to the induction of acid resistance in *Salmonella* Enteritidis (*S.* Enteritidis) in laboratory media. The second part of this study aimed to evaluate whether the results from the broth media could be extrapolated in foods, particularly in an acid food matrix (tarama salad containing citric acid as acidulant), stored under refrigeration.

## Materials and methods

### Bacteria strain and growth conditions

*Salmonella enterica* ssp. *enterica* serovar Enteritidis (*S.* Enteritidis) P167807 Phage Type 4 (PT4), a food (beef) isolate reported in Boziaris et al. [23] was provided by the Laboratory of Food Microbiology and Biotechnology, Agricultural University of Athens, Greece.

Cells were monthly subcultured in Tryptone Soy Agar (TSA, Lab M Limited, Lancashire, UK) from stock cultures (-20˚C) and maintained at 4˚C. Prior to each experimentation, one single colony was transferred to 10 ml of Tryptone Soy Broth without dextrose (TSBG(-), Lab M Limited, Lancashire, UK) and incubated at 37˚C for 24 h. Subsequently, 100 μl of the 24-h cultures were transferred to 10 ml of the same medium and incubated at 37˚C for another 18 h, in order to collect stationary phase cells.

## Minimum Inhibitory Concentration (MIC) determination

Stationary phase cultures were washed twice with ¼ Ringer solution (Lab M Limited, Lancashire, UK) (2709 X g, 10min, 4˚C) and resuspended in the appropriate medium. For the preparation of the media, Tryptone Soy Broth (Lab M Limited, Lancashire, UK) was supplemented with several concentrations (10, 20, 30, 40, 50, 75, 100, 150 and 200 mM) of acetic acid (Panreac, Barcelona, Spain) and then the pH was adjusted to 5.0 using HCl 6 N (Merck, Darmstadt, Germany) or NaOH 10 N (Panreac, Barcelona, Spain). After autoclave, the pH value of each acid concentration was confirmed with a digital pH-meter (pH 526, Metrohm Ltd, Switzerland) and differences (maximum ± 0.2) -if evident- were taken into consideration in the final assay. Samples were inoculated with approximately 5.0 log CFU/ml and incubated at 37˚C for up to 10 days to assess the growth responses. Sampling was performed on day 0 and after 5 and 10 days of storage by plating 0.1 ml of the appropriate dilution on TSA plates. The experiment was conducted four independent times with duplicate samples per trial.

## Preparation and inoculation of adaptation media and acid challenge assays

Adaptation media were prepared by combining different concentrations of total acetic acid and pH values. More specifically, appropriate volumes of acetic acid (1 M) were added to 100 ml of TSBG(-). The media were then autoclaved, adjusted to the desired pH values with HCl 6 N or NaOH 10 N, in order to create different concentrations of undissociated acetic acid (UAA) and filtered-sterilized (0.2 μm, LLG Labware, USA). In all assays, adaptation was performed for 90 minutes at 37˚C to a preheated water bath. Enumeration of the initial adapted populations was carried out at the end of adaptation period by transferring 100 μl in 900 μl of ¼ Ringer and plating the appropriate dilution on TSA plates. Non-adapted (NA) cells grown at neutral pH (7.00) without being subjected to any pH adjustment or acetic acid pre-exposure were also used as controls in all experiments.

**Adaptation and exposure to TSB adjusted to pH 2.5 (screening assay).** For screening assay, four different concentrations of total acetic acid (15, 25, 35 and 45 mM) were combined with pH adjusted to 4.0, 4.5, 5.0, 5.5 and 6.0 (± 0.05), as described above. Cells adapted to pH in the absence of acetic acid (pH-adapted cells; 0mM acetic acid) were used as 'positive' controls by adjusting the pH of the broth medium to the same values as those mentioned above using only HCl 6 N. For the preparation of adapted cultures, stationary phase cells were centrifuged (2709 X g, 10 min, room temperature) and resuspended to the appropriate adaptation medium at a final concentration of approximately 6.5–7.0 log CFU/ml. Following adaptation, cells were harvested by centrifugation (2709 X g, 5 min, 37˚C) and immediately resuspended to TSB adjusted to pH 2.5 with HCl 6 N (TSB$_{2.5}$) at a final concentration of ~ 5.0 CFU/ml. Non-adapted (control) inocula were resuspended to TSB$_{2.5}$ without prior adaptation. Acid challenge was performed at 37˚C for 30 minutes. Samplings were carried out at 0, 2.5, 5 and 7.5 minutes of acid exposure by plating 0.1 ml of the appropriate dilution on TSA (detection limit of 1.0 log CFU/ml) and at 10, 15 and 30 minutes of exposure, by plating 1 ml of the challenged broth into three TSA petri dishes (detection limit of 0 log CFU/ml). Experiments were conducted in triplicate with duplicate samples per independent trial.

**Impact of adaptive responses in tarama salad or TSB adjusted to pH 4.35.** Based on the results of the screening assay (TSB$_{2.5}$), three acetic acid/pH combinations i.e. 15mM/pH5.0, 35mM/pH5.5 and 45mM/pH6.0, were selected. These treatments were further tested for their ability to induce acid resistance of *S*. Enteritidis at commercial tarama salad stored at 5°C and at TSB adjusted to pH 4.35 (TSB$_{4.35}$) with HCl 6 N or citric acid 6 M (AnalaR, Dublin, Ireland) and incubated at 37°C or 5°C. Exposure of *S*. Enteritidis to TSB$_{4.35}$ incubated at two temperatures was performed in order to isolate the impact of food matrix on pathogen survival at the same pH value, acidulant and challenge temperature.

Preparation of adapted cultures inoculated in broth medium (TSB$_{4.35}$) was performed as described above. Briefly, stationary phase cells were centrifuged (2709 X g, 10 min, room temperature) and resuspended to each of the above adaptation media (i.e. 15mM/pH5.0, 35mM/pH5.5 or 45mM/pH6.0) at a final concentration of approximately 6.5–7.0 log CFU/ml. Following adaptation, cells were harvested by centrifugation (2709 X g, 5 min, 37°C) and resuspended to TSB$_{4.35}$ adjusted either with HCl (to stimulate food matrix pH) or citric acid (to stimulate food matrix acidulant), at a final concentration of ~ 5.0 CFU/ml. Non-adapted (control) inocula were resuspended to TSB$_{4.35}$ without prior adaptation. Samples were stored at 5°C for 60 days or at 37°C for 96 hours (when citric acid was used for pH adjustment) or 7 days (when HCl was used for pH adjustment). Samplings were performed at different time intervals, depending on storage temperature and the use of HCl or citric acid for lowering the pH. Enumeration was carried out by plating 0.1 ml of the appropriate dilution on TSA plates containing 0.1% sodium pyruvate (Applichem, Darmstadt, Germany) (TSA/SP) (detection limit of 1.3 log CFU/ml). This medium was selected for enabling the maximum recovery of injured cells [24]. Experiments were conducted four independent times with duplicate samples per replicate.

For the inactivation experiments in the acid food matrix, commercial tarama salad packages of a Greek food industry were purchased from a local supermarket and transferred to the laboratory within 20 minutes. Tarama salad is a traditional Greek fish roe appetizer (spread) stored under refrigeration. Acidification is performed using citric acid. Apart from fish roe and citric acid, other ingredients used for the preparation of the product according to the labelling were mashed potatoes, vegetative oil, salt, pigments, flavorings, condenser and chemical preservative (sodium benzoate, sorbic acid). This product was selected since it is a domestically widespread acid food with very low to undetectable initial microbial load. Prior to inoculation, levels of indigenous microbiota of commercial packages was determined by diluting 10 g of each uninoculated package to 90 ml of ¼ Ringer and plated on TSA/SP plates. The pH of commercial tarama salad was also measured using a digital pHmeter.

For the preparation of adapted cultures inoculated in tarama salad, stationary phase cells were centrifuged (2709 X g, 10 min, 4°C), washed twice with ¼ Ringer and then resuspended to each of the above adaptation media (i.e 15mM/pH5.0, 35mM/pH5.5 or 45mM/pH6.0) at a final concentration of approximately 8.5–9.0 log CFU/ml. Following adaptation, cells were harvested by centrifugation (2709 X g, 5 min, 37°C) and resuspended to 4 ml of diluted tarama salad prepared by mixing10 g of tarama salad with 30 ml ¼ Ringer. This was performed in order to acclimatize inocula in a medium similar to the subsequent food substrate. The suspension was vortexed for 1 minute and aliquots (0.8 ml) were added to 80 g of commercial tarama salad pre-weighed in sterilized containers. Non-adapted inocula were resuspended to 4 ml of diluted tarama salad without prior adaptation. The initial inoculated population was approximately 6.5–7.0 log CFU/g. Samples were stored at 5°C for 37 days. Samplings were conducted by transferring 10 g of inoculated tarama salad in 90 ml ¼ Ringer solution and homogenized in a stomacher apparatus (Seward, London, UK). Then, 0.1 ml of the appropriate dilution was spread on TSA/SP plates (detection limit of 2 log CFU/g). pH changes during storage were

determined using a digital pHmeter. Experiments were conducted four independent times with duplicate samples per replicate.

## Determination of 4D inactivation parameters

The time needed for a 4 log reduction (4D) of the microbial population was calculated by fitting the log transformed inactivation data collected from the screening assay at $TSB_{2.5}$ to Weibull with tail (Albert) model according to the equation $\log10(N) = \log10[(N(0)-N_{res}) \times 10^{(-(t/\delta)^p)} + N_{res}]$ [25], whereas the log transformed data of tarama salad inactivation curves were fitted in Weibull model according to the equation $\log10(N) = \log10(N(0))-(t/\delta)^p$ [26], where $N_o$ the population at time $t_o$, N the population at time t, $N_{res}$ the residual bacterial concentration log CFU/g at the end of microbial inactivation, $\delta$ the time needed for the first decimal reduction and $p$ a shape parameter corresponding to different concavities; downward concave survival curves for p>1, upward concave survival curves for p<1 or linear curves for p = 1. GinaFit, a freeware Add-in for Microsoft ®Excel [27] available at https://cit.kuleuven.be/biotec/software/GinaFit was used for data fitting. In total, six curves per experimental case were fitted.

## Statistical analysis

Log transformed inactivation data and 4D inactivation estimates were used for statistical analysis. Analysis of variance (SPSS 22.0 for Mac) was performed among cell populations (log CFU/ml or g) for each time point during all stresses and among 4D values calculated from the screening assay ($TSB_{2.5}$) and 'tarama' salad inactivation. Means were compared using Tukey's Honestly Significant Difference (HSD) test and were considered significant at 95% level. Comparison between NA and pH-adapted cells was performed using *t-test* of Microsoft® Excel 16 for Mac.

# Results

## MIC determination

The MIC of UAA for *S*. Enteritidis ranged between 5.2 and 7.2 mM. At concentrations of UAA $\leq$ 5.2 mM growth was observed at the 5th day of storage, while concentrations $\geq$ 7.2 mM displayed a bactericidal effect.

## Screening assay at $TSB_{2.5}$

Populations at the end of adaptation period (90 min) ranged from 6.1 to 6.9 CFU/ml. The effect of adaptation to pH (i.e., 0mM/pH 6.0, 5.5, 5.0, 4.5 and 4.0; no added acetic acid) on *S*. Enteritidis acid resistance was evaluated comparing bacterial populations of pH-adapted and NA inocula (Fig 1). Adaptation to pH 6.0 did not increase acid resistance, since no marked differences (P>0.05) were observed during 2.5–15 minutes of acid exposure. In contrast, significantly higher (P<0.05) survivors were recovered at the lower pH values 4.0–5.5 compared to NA cells. Nonetheless, the magnitude of the observed differences was pH-dependent. Log differences following adaptation to pH 5.5 and 5.0, albeit statistically significant (P<0.05), were rather low (0.7–1.4 log CFU/ml). Adaptation to pH 4.5 and 4.0, on the other hand, clearly enhanced resistance: 1.3–2.5 and 0.6–1.9 log CFU/ml higher survivors were enumerated following adaptation to pH 4.5 and 4.0, respectively, compared to control (NA) (Fig 1). The ratio of $4D_{pH}/4D_{NA}$ (Fig 2) was used to characterize the pH induced acid resistance, where $4D_{pH}$ and $4D_{NA}$ the 4D parameters of the pH adapted and NA inocula, respectively, calculated by Weibull with tail (Albert) model. The ratio was calculated by dividing $4D_{pH}$ of each replicate with an average value of $4D_{NA}$. A clear difference (P<0.05) was observed between treatments adapted to pH 6.0 and 4.5, with higher values, i.e., suggesting longer survival, obtained at the

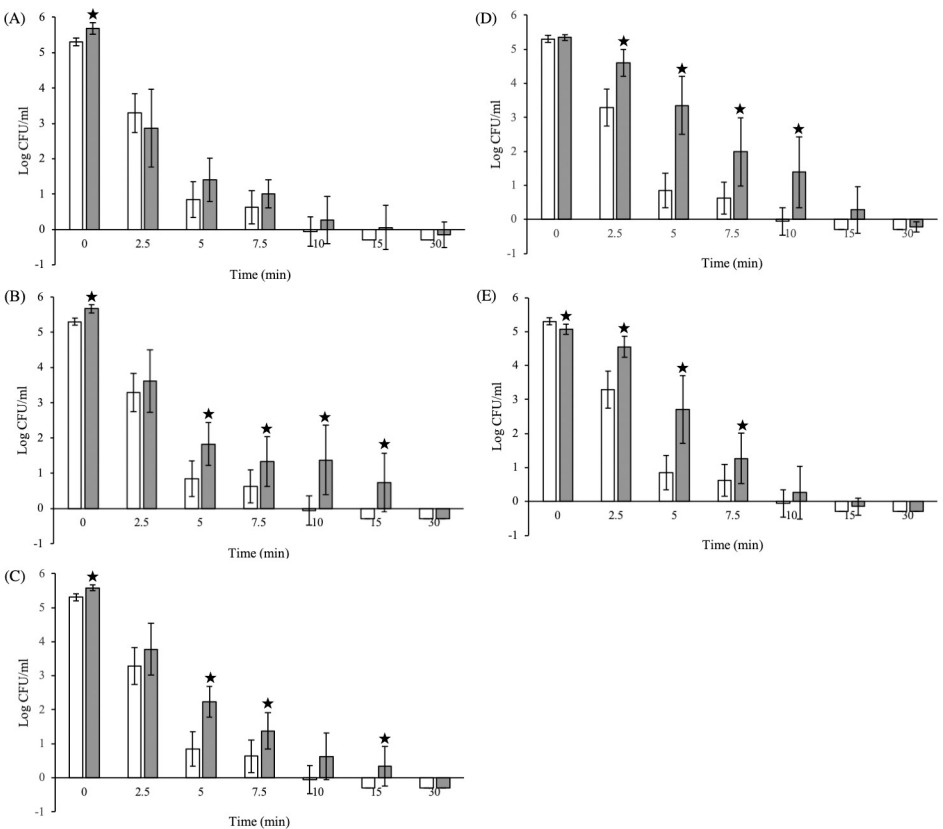

**Fig 1. Inactivation of NA and pH-adapted cells of *S.* Enteritidis at pH 2.5 (TSB$_{2.5}$).** Adaptation was performed to pH values (A) 6.0, (B) 5.5, (C) 5.0, (D) 4.5 and (E) 4.0. Reduction of the pH of the adaptation medium from 6.0 to 4.0 increased the enumerated survivors compared to control (NA), with maximum differences found for cells adapted to pH 4.0 and 4.5. White and grey bars represent NA and adapted populations, respectively. Each bar is an average of six replicates (± standard deviation). Stars indicate significant differences between two treatments for each time point according to *t-test*.

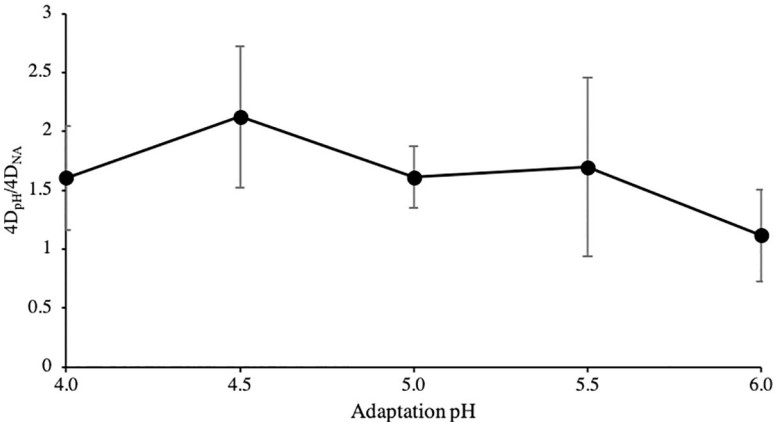

**Fig 2. Induced acid resistance of *S.* Enteritidis following adaptation to different pH values.** Each data point represents a mean ratio $4D_{pH}/4D_{NA}$ (± standard deviation), where $4D_{pH}$ and $4D_{NA}$ the 4D's for pH adapted and NA inocula, respectively, resulting from diving $4D_{pH}$ of each replicate with an average value of $4D_{NA}$. Lower pH value 4.5 induced a higher ratio and, thus, an increased survival compared to higher pH value 6.0.

lower pH (4.5). In summary, adaptation over a range of different pH values enhanced survival, with maximum acid resistance induced by lower pH values.

Adaptation to acetic acid enhanced acid resistance in a pH- and UAA concentration-dependent manner. Regarding cultures adapted to pH 6.0, a gradual increase in the ability of the pathogen to endure severe acid stress was observed with increasing concentrations of UAA. Adding 1.9 (35mM/pH6.0) and 2.4 (45mM/pH6.0) mM UAA acid resulted in up to 1.5–2.0 log units (P<0.05) higher counts compared to cultures adapted to treatments with lower levels or without UAA (i.e., 0, 0.8 and 1.4 mM corresponding to 0mM/pH6.0, 15mM/pH6.0 and 25mM/pH6.0, respectively) (Fig 3A; Table 1). This trend was also confirmed by comparing the inactivation kinetics (Table 1). The addition of 1.9 (35mM/pH6.0) and 2.4 (45mM/pH6.0) mM UAA increased the time needed for a four-log reduction (4D) by more than two-fold, i.e. from 5.0 minutes required for control cells (0mM/pH6.0) to 10 (P<0.05) and 11.6 min (P<0.05), respectively (Table 1).

A similar trend was found for cells adapted to treatments with a lower final pH 5.5, though higher amounts of UAA were required to induce acid resistance at pH 5.5 compared to pH 6.0. For instance, adding 2.3 mM UAA (15mM/pH5.5), a similar concentration of UAA as in the case of 45mM/pH6.0, as well as 3.8 mM UAA (25mM/pH5.5) resulted in 1.0–1.7 log units

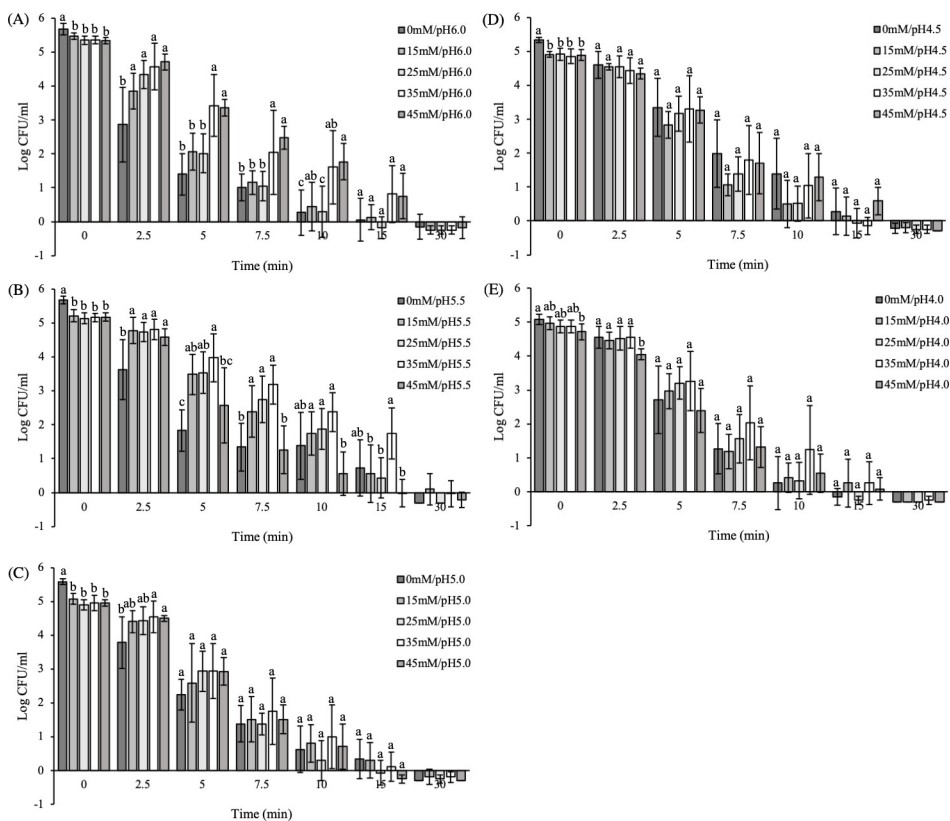

**Fig 3. Inactivation of adapted S. Enteritidis cells at pH 2.5 (TSB$_{2.5}$) (screening assay).** Diagrams represent treatments adapted to pH (A) 6.0, (B) 5.5, (C) 5.0, (D) 4.5 and (E) 4.0, whereas bars within each diagram different (undissociated) acetic acid concentrations. Adaptation to pH 6.0 and 5.5 within a range of increasing concentrations of UAA (1.9–5.4 mM) induced elevated acid resistance compared to the individual effect of pH. Adaptation to lower pH values (5.0, 4.5 and 4.0) had no effect, regardless of the amount of UAA. Each bar is an average of six replicates (± standard deviation). Different letters indicate significant differences among treatments of the same time interval according to Tukey's HSD test.

**Table 1. Adaptive treatments without (pH adapted cells) or in the presence of UAA used in the current study and their kinetic parameter estimates during acid inactivation at TSB$_{2.5}$.**

| pH adjusted with HCl/NaOH | Total AA[a] (mM) | UAA[b] (mM) | 4D | RMSE[c] | R[2] [d] |
|---|---|---|---|---|---|
| pH 6.0 | 0 | 0.0 | 5.0 ± 1.7 (a) | 0.2092–0.6124 | 0.9788 ± 0.0116 |
| | 15 | 0.8 | 6.7 ± 1.3 (ab) | 0.1499–0.522 | 0.9787 ± 0.0177 |
| | 25 | 1.4 | 6.4 ± 1.4 (ab) | 0.2746–0.5035 | 0.9850 ± 0.0060 |
| | 35 | 1.9 | 10.0 ± 3.8 (bc) | 0.1742–0.4406 | 0.9840 ± 0.0086 |
| | 45 | 2.4 | 11.6 ± 2.1 (c) | 0.1029–0.3306 | 0.9903 ± 0.0082 |
| pH 5.5 | 0 | 0.0 | 7.6 ± 3.5 (a) | 0.2046–0.7463 | 0.9567 ± 0.0309 |
| | 15 | 2.3 | 10.9 ± 2.8 (a) | 0.2023–0.3894 | 0.9823 ± 0.0076 |
| | 25 | 3.8 | 11.2 ± 1.5 (a) | 0.1239–0.4631 | 0.9843 ± 0.0151 |
| | 35 | 5.4 | 16.0 ± 3.7 (b) | 0.1603–0.3591 | 0.9827 ± 0.085 |
| | 45 | 6.9 | 7.7 ± 1.9 (a) | 0.1824–0.6411 | 0.9830 ± 0.0128 |
| pH 5.0 | 0 | 0.0 | 7.2 ± 1.2 (a) | 0.0718–0.5381 | 0.9782 ± 0.0187 |
| | 15 | 5.5 | 8.4 ± 1.7 (a) | 0.1628–0.6340 | 0.9701 ± 0.0215 |
| | 25 | 9.1 | 8.4 ± 0.6 (a) | 0.0974–0.5299 | 0.9842 ± 0.0156 |
| | 35 | 12.8 | 8.9 ±2.3 (a) | 0.1442–0.6641 | 0.9801 ± 0.0130 |
| | 45 | 16.4 | 8.9 ±1.0 (a) | 0.1716–0.5709 | 0.9826 ± 0.0156 |
| pH 4.5 | 0 | 0.0 | 9.7 ± 2.8 (a) | 0.1087–0.6569 | 0.9873 ± 0.0199 |
| | 15 | 9.7 | 7.5 ±0.7 (a) | 0.2253–0.5100 | 0.9735 ± 0.0201 |
| | 25 | 16.1 | 8.5 ± 0.7 (a) | 0.1204–0.5669 | 0.9798 ± 0.0236 |
| | 35 | 22.6 | 9.2 ± 1.7 (a) | 0.1613–0.456 | 0.9877 ± 0.0102 |
| | 45 | 29.0 | 10.8 ± 2.8 (a) | 0.1876–0.3979 | 0.9819 ± 0.0101 |
| pH 4.0 | 0 | 0.0 | 7.2 ± 2.0 (a) | 0.1821–0.4894 | 0.9786 ± 0.0112 |
| | 15 | 12.8 | 7.9 ± 0.9 (a) | 0.2534–0.6946 | 0.9758 ± 0.0250 |
| | 25 | 21.3 | 8.4 ± 1.1 (a) | 0.0855–0.4374 | 0.9923 ±0.0078 |
| | 35 | 29.8 | 10.1 ± 3.1 (a) | 0.2043–0.4317 | 0.9856 ± 0.0091 |
| | 45 | 38.3 | 8.6 ± 1.6 (a) | 0.1485–0.3170 | 0.9831 ± 0.0187 |

Weibull with tail (Albert) model was used for calculation of 4D, RMSE and R$^2$ inactivation estimates. Values represent mean (± standard deviation) of six replicates. Different letters among treatments adapted at a given pH indicate significant differences of 4D values according to Tukey's HSD test.

[a] AA: Acetic acid.

[b] Theoretical undissociated acetic acid was calculated according to Henderson–Hasselbalch equation.

[c] RMSE: Root Mean Square Error.

[d] R$^2$: regression coefficient.

(P<0.05) higher survivors compared to the individual effect of pH 5.5 (no added UAA, 0mM/pH5.5), but did not have any impact on the 4D estimates (P>0.05) (Fig 3B, Table 1). Therefore, the log differences were not taken into consideration, since they were considered marginal. An additional increase in UAA concentrations to 5.4 mM (35mM/pH5.5) resulted in up to 2.2 log CFU/ml higher survivors compared to control (0mM/pH5.5) (Fig 3B) and the highest (P<0.05) 4D value (16.0 min; Table 1). Further increase of UAA (6.9 mM, 45mM/pH5.5) limited survival and decreased 4D values to similar levels as in control (0mM/pH5.5) (Fig 3B, Table 1).

Contrary to the above results, no effect on the acid resistance of *S.* Enteritidis was observed when cells were previously exposed to acetic acid at pH values equal to or lower than 5.0 (Fig 3C–3E, Table 1). This was evident even in concentrations of UAA that induced acid resistance at higher pH value (i.e. 5.5). For these treatments, similar log counts (P>0.05) and 4D estimates (P>0.05) were obtained for all UAA concentrations (Fig 3C–3E, Table 1).

Overall, the induced acid resistance of *S*. Enteritidis PT4 against subsequent acid exposure to pH 2.5 (TSB$_{2.5}$) was affected by both pH and UAA concentration of the adaptation medium. Adaptation to pH 6.0 and 5.5 within a range of increasing concentrations of UAA (1.9–5.4 mM) successfully protected pathogen against the subsequent severe acid stress.

## Exposure to 'tarama' salad

In order to examine whether the results from the laboratory media could be extrapolated in an acid food matrix stored under refrigeration, three adaptation treatments (15mM/pH5.0, 35mM/pH5.5 and 45mM/pH6.0) were selected for culture preparation prior to inoculation in tarama salad. The selection of the treatments was based on the concentration of UAA and their adaptive responses at TSB$_{2.5}$, as follows: 45mM/pH6.0 and 35mM/pH5.5 had different concentrations of theoretical UAA (2.4 mM and 5.4 mM, respectively) (Table 1) but both increased acid resistance of *S*. Enteritidis (Fig 3A and 3B). On the other hand, 15mM/pH5.0 contained approximately the same amount of UAA as 35mM/pH5.5 (5.5 and 5.4 mM, respectively) (Table 1), but had no effect on the subsequent acid resistance of the pathogen (Fig 3B and 3C). Non-adapted inocula were also used as control.

A total of 21 commercial tarama salad packages were used throughout the study. The initial pH of the salad was 4.35 ± 0.02. The levels of indigenous microflora were not quantifiable in the packages tested. No marked differences (P>0.05) were observed among enumerated populations and 4D values following inoculation in tarama salad at 5°C, irrespectively of the preceded adaptation treatment or the control (NA) (Fig 4, Table 2). This is in contrast to the results of the screening assay, where adaptation to 35mM/pH5.5 and 45mM/pH6.0 adequately strengthened the cells against the subsequent severe acid stress (Fig 3A and 3B). pH values remained unchanged throughout the storage period (P>0.05) (S1 Fig).

## Exposure to pH 4.35 using HCl and citric acid

Since acid adaptive responses following exposure to TSB$_{2.5}$ at 37°C were different from those obtained in tarama salad stored under refrigeration (5°C), an effort was made to examine the reasons underpinning these discrepancies. Thus, the effect of key individual factors for *S*. Enteritidis inactivation that may be of relevance to an acid food matrix (e.g., pH value, acidulant agent) and the effect of challenge temperature (5°C, 37°C) was further examined. Cells

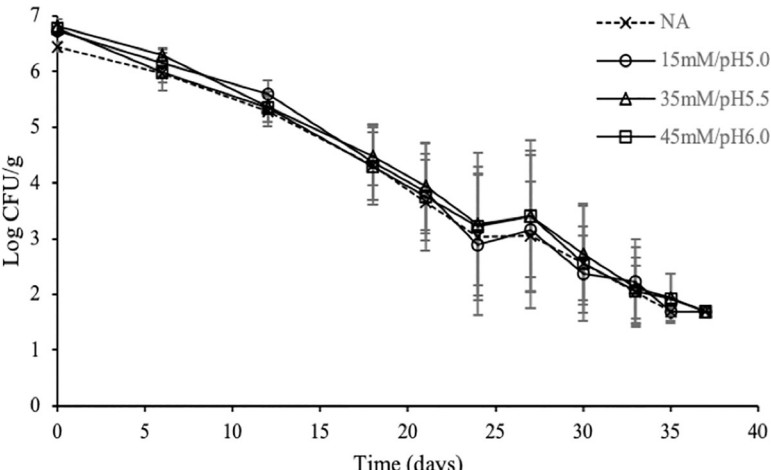

**Fig 4. Acid adapted *S*. Enteritidis cells were not protected against tarama salad stored at 5°C.** Each data point represents an average of six to eight replicates (± standard deviation).

**Table 2. Kinetic parameter estimates of Weibull model during inactivation in tarama salad.**

| Treatment | 4D | RMSE[a] | $R^{2}$ [b] |
|---|---|---|---|
| NA | 28.0 3 ± 5.32 **(a)** | 0.0607–0.3426 | 0.9838 ± 0.0109 |
| 15mM/pH5.0 | 27.16.7 ± 5.6 **(a)** | 0.0855–0.5095 | 0.9760 ± 0.0144 |
| 35mM/pH5.5 | 28.1± 6.0 **(a)** | 0.1996–0.3776 | 0.9779 ± 0.0070 |
| 45mM/pH6.0 | 28.75 ± 6.12 **(a)** | 0.1721–0.3650 | 0.9758 ± 0.0161 |

Values represent mean (± standard deviation) of six to eight replicates. Different letters indicate significant differences among 4D values according to Tukey's HSD test.

[a] RMSE: Root Mean Square Error.

[b] $R^{2}$: regression coefficient.

were first adapted to the selected treatments (15mM/pH5.0, 35mM/pH5.5 or 45mM/pH6.0), inoculated in $TSB_{4.35}$ (pH of the commercial tarama salad) using either HCl or citric acid (acidulant of tarama salad) and incubated at 37 or 5°C (temperature effect).

Adaptation to 35mM/pH5.5 failed to protect the pathogen against the subsequent lethal stress of pH 4.35 ($TSB_{4.35}$) at 37 and 5°C, regardless of the acidulant used (Figs 5 and 6). Even though this observation is not in line with the results obtained following $TSB_{2.5}$ inactivation at 37°C (Fig 3B), it may explain the inability of these culture types to tolerate the lethal environment of tarama salad stored at 5°C (Fig 4). On the other hand, adaptation to 45mM/pH6.0 successfully prolonged the survival of pathogen in $TSB_{4.35}$ at 37°C for both acidulants used (Fig 5, S3 Table). The ability of 45mM/pH6.0 adapted cells to survive lethal acid stress at 37°C was also observed following exposure to $TSB_{2.5}$ (Fig 3A), though this trend was not confirmed in tarama salad stored at 5°C (Fig 4). More specifically, exposure to $TSB_{4.35}$ adjusted with citric acid following adaptation to 45mM/pH6.0 resulted in up to 1.8 log units (P<0.05) higher populations during 32–96 hours of incubation compared to the rest treated and untreated cultures (i.e., control and 15mM/pH5.0, 35mM/pH5.5) (Fig 5A). Exposure to HCl at 37°C had a bactericidal effect for cells adapted to 15mM/pH5.0 and 35mM/pH5.5 as well as NA inocula, reducing the microbial load until the 5-7th day of storage (Fig 5B). Nonetheless, cells adapted to 45mM/pH6.0 exhibited a low reduction up to 1 log CFU/ml followed by an increase to their initial levels at the 3rd day of storage (S3 Table). On the contrary, a shift in the incubation temperature from 37°C to 5°C sensitized 45mM/pH6.0 inocula (1.0–1.5 log CFU/ml lower counts) (P<0.05) against the subsequent acid and cold stress compared to 15mM/pH5.0, 35mM/pH5.5 and control (NA) cells (Fig 6).

In line with the results from the screening assay and tarama salad inactivation, inoculation of 15mM/pH5.0 adapted cells in $TSB_{4.35}$ did not affect their survival, irrespectively of the incubation temperature or the acidulant used (Figs 5 and 6).

Overall, the pH value of the challenge medium as well as the incubation (challenge) temperature determined the responses of *S*. Enteritidis following adaptation to 35mM/pH5.5 and 45mM/pH6.0, respectively, but not to 15mM/pH5.0. On the contrary, similar trends were found when different acidulants were used for lowering the pH.

## Discussion

In the present study, the effect of adaptation to UAA and pH on the subsequent acid adaptive responses of *S*. Enteritidis was examined in laboratory media and in an acid food matrix. Acetic acid was used as the adaptation agent, since it is the predominant acid in many foods, such as mayonnaise, salad dressings and sauces [14]. The amount of undissociated acetic acid in the adaptation treatments was estimated according to Henderson-Hasselbalch equation. Experimental design covered a range of different combinations of pH values and concentrations of

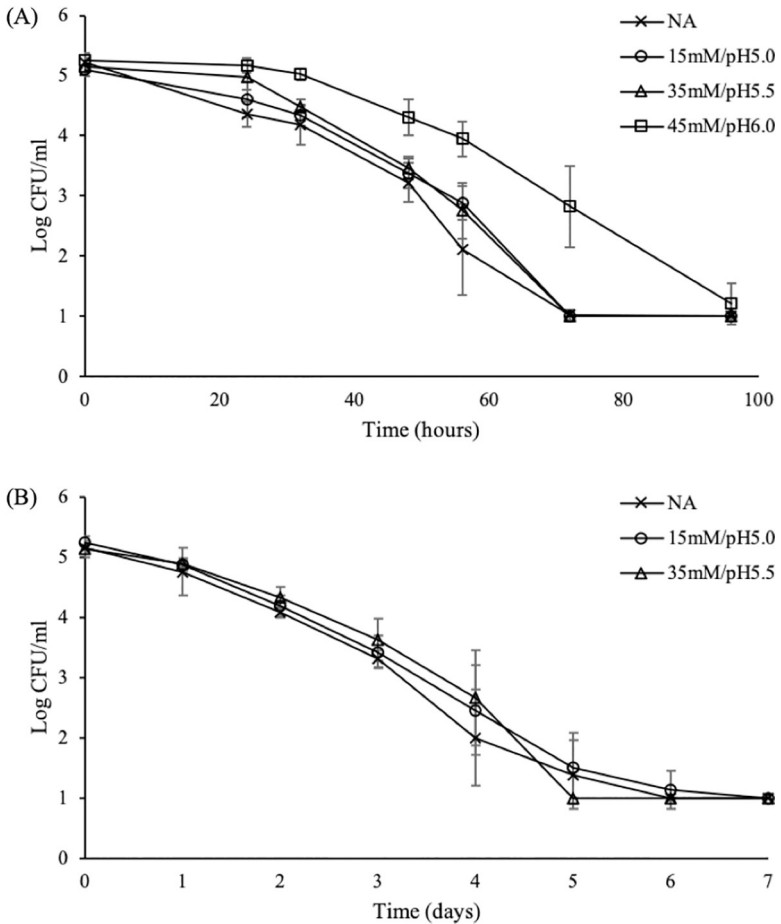

**Fig 5. Effect of food matrix (pH, acidulant) on the responses of S. Enteritidis at 37˚C.** Cells were exposed to $TSB_{4.35}$ at 37˚C adjusted either with citric acid (A) or HCl (B) following adaptation to 15mM/pH5.0, 35mM/pH5.5 and 45mM/pH6.0. Each data point is an average of eight replicates (± standard deviation).

acetic acid *Salmonella* spp. may encounter in food-related ecosystems. *Salmonella* Enteritidis serotype was chosen as it was the most prevalent in Europe [28].

The individual effect of pH in the induced acid resistance of the pathogen was examined by adjusting the pH of the adaptation medium (TSBG(-)) with hydrochloric acid. This inorganic acid was used because it is completely dissociated in aqueous environments and therefore itself is not toxic for the cells [29]. In line with previous reports, adaptation to moderate pH values (4.0–5.5) protected cells against subsequent acid exposure, with higher resistance observed at the lower pH values (4.0 and 4.5) [30–33]. Nonetheless, the pH values required for the induction of acid resistance in pH adapted cells were lower compared to the milder pH values (5.5–6.0) required in the presence of acetic acid.

The addition of acetic acid to the adaptation media also enhanced survival of the pathogen following adaptation to some of the treatments tested. In general, the ability of weak organic acids to activate acid resistance mechanisms has previously been reported [9–13,34–37]. Adaptation to juices, natural sources of organic acids, was also found to elicit a protective effect in *Salmonella* spp. and *E. coli* [38,39]. According to Yuk and Marshall [35] and Yuk and Schneider [38], the differences observed in the phenotypic responses of individual strains of *E. coli* and *Salmonella* spp. adapted to organic acids and juices, respectively, can be ascribed to the

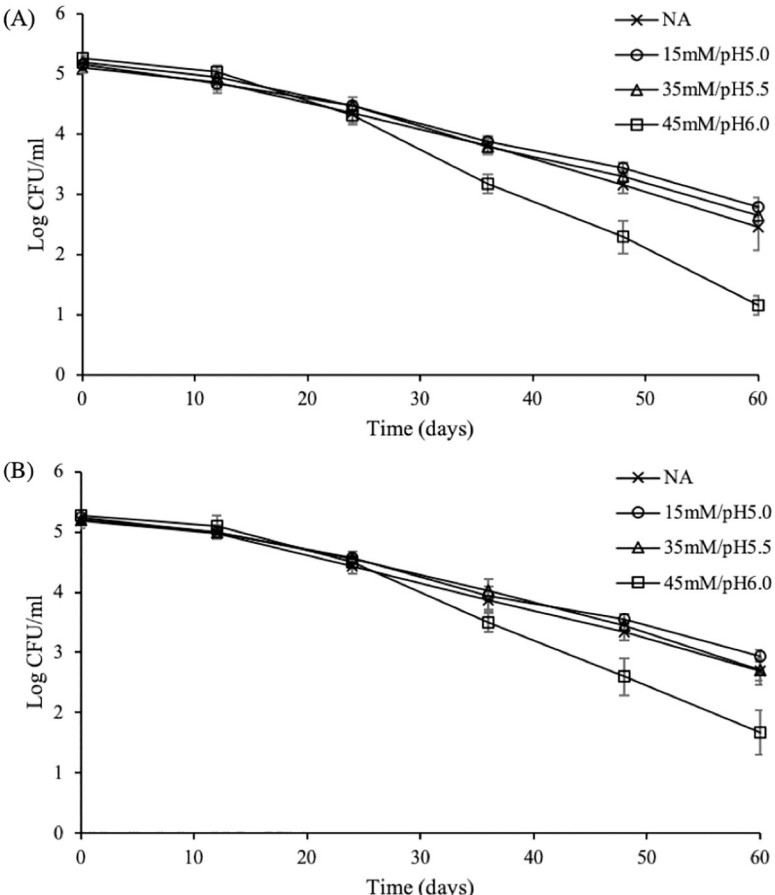

**Fig 6. Effect of food matrix (pH, acidulant) and refrigeration temperature (5˚C) on the responses of *S*. Enteritidis.** Cells were exposed to TSB$_{4.35}$ at 5˚C adjusted either with citric acid (A) or HCl (B) following adaptation to 15mM/pH5.0, 35mM/pH5.5 and 45mM/pH6.0. Adaptation to 45mM/pH6.0 sensitized pathogen against the subsequent acid and cold stress. Each data point is an average of eight replicates (± standard deviation).

different amounts of undissociated acid present in the adaptation media. Acetate has been found to activate *rpoS* in log phase cells [40]. This gene is the master regulator of the general stress response in many Gram⁻ bacteria [41] induced upon entry of cells into stationary phase of growth and under a variety of unfavorable conditions, such as low pH [42]. Lee et al. [43] has reported that induction of *rpoS* by acetate is performed indirectly, through the reduction of intracellular pH.

Nevertheless, the impact of different levels of undissociated acid in the induced acid resistance of *Salmonella* has not been widely investigated. Therefore, the present study tries to establish a link between the levels of theoretical UAA calculated according to Henderson-Hasselbalch equation and the induced acid resistance of *S*. Enteritidis PT4 under extreme acid conditions. Based on the results of the broth screening, previous exposure of the pathogen to acetic acid increased its acid resistance in a UAA concentration- and pH-dependent manner. In mild pH values (5.5–6.0), increasing the amount of UAA within a range (1.9–5.4 mM) protected the pathogen against the detrimental effect of severe pH (2.5), even though higher concentrations of UAA were required at pH 5.5 compared to pH 6.0 for the induced resistance to be manifested. The concentrations of UAA that induced acid resistance were in each case lower, or at least close to the lowest limit of the calculated MIC range (5.2–7.2 mM), indicating

that levels below the growth/no growth limit were needed for the stimulation of the protective effect. Adaptation to UAA close to the upper limit of the MIC (6.9 mM UAA, treatment 45mM/pH5.5) had no effect on the acid tolerance of *S*. Enteritidis. On the other hand, addition of UAA in lower pH values (5.0–4.0), where a higher concentration of protons was present, had no effect on the acid phenotypic responses, regardless of the concentrations of theoretical UAA that were added.

Alterations in factors prevailing in the challenge substrate or during challenge drastically affected the ability of the pathogen to tolerate stress in culture media. Interestingly, increasing the pH of the challenge medium (TSB) from 2.5 to 4.35 clearly suppressed the induced resistance of cells adapted to 35mM/pH5.5 at both incubation temperatures (5 and 37˚C) and acidulants (HCl, citric acid) tested. Although the exact reason for this shift is not known, different mechanisms triggered to support survival under extreme acid conditions may be significantly affected by conditions prevailing at the challenge substrate. For instance, the so-called Acid Tolerance Response (ATR) may protect stationary phase cells at external pH 3.0 [30], but will not provide significant protection at pH 2.5 [44]. Amino acids-dependent pH homeostatic mechanisms, on the other hand, may enhance resistance at pH 2.5 [45] or pH 2.3 [46], provided that the cognate amino acids are available in the substrate. TSB used in the broth experiments is a rich medium, containing tryptone and soy peptone as protein sources, therefore providing the necessary amino acids. Contrary to the responses of 35mM/pH5.5 adapted cells, acid resistance of 45mM/pH6.0 inocula adapted to lower concentrations of UAA, but higher pH was affected only by incubation temperature. These inocula were sensitized when exposed to pH 4.35 (TSB$_{5.35}$) at 5˚C but not at 37˚C, regardless of the acidulant used. This result is in line with those reported from Tiwari et al. [47], who also found that prolonged exposure of acid adapted *Salmonella* cells to acidic conditions at 4˚C was more detrimental (e.g. had higher reductions) compared to the control samples stored at pH 7.3 at the same temperature. Exposure of *S*. Seftenberg non-adapted and acid-adapted cells to acid and cold stress altered the membrane fatty acid composition of both inocula, with higher changes found in acid adapted cultures. These alterations resulted in increased bacterial membrane fluidity [48]. It is generally believed that a lower membrane fluidity correlates well with higher acid resistance of bacterial cells [32, 35, 49]. As demonstrated before, a shift in the storage temperature can affect the acid resistance phenotypes. Shen, Yu and Chou [50] reported that whereas no differences were observed between acid adapted and non-adapted cells of *S*. Typhimurium inoculated in skim milk and treated fermented milk stored at 37˚C, acid adaptation, in addition to promoting acid resistance, decreased the susceptibility of the pathogen to refrigeration (5˚C). Nonetheless, it is not the first time that the effects of acid adaptation are counteracted by subsequent stressors, increasing the sensitivity of acid adapted cells to lethal stresses compared to their non-adapted counterparts [33,51,52].

When tarama salad was used instead of TSB$_{4.35}$ at 5˚C, survival of all adapted cells was similar to the non-adapted cultures. Notably, enhanced resistance of acid adapted *S*. Typimurium was reported in fermented milk products stored at 5˚C [50]. The increased sensitivity of 45mM/pH6.0 adapted cells at TSB$_{4.35}$ at 5˚C but not at tarama salad at 5˚C suggests a protective effect of food matrix on these inocula. This result indicates that intrinsic factors of tarama salad other than the pH value and the acidulant *per se* may also determine the resistance phenotypes. It has been found that a combination of organic acids and NaCl may elicit a protective effect against *Salmonella*, *E. coli* and *Shigella flexneri* [53–55], principally due to a raise in the intracellular pH [55]. In addition, certain food ecosystems may help bacterial cells tolerate lethal acid environments [12,22]. For instance, Alvarez et al. [12] reported that acid adapted cells challenged to Meat Extract at pH 3.0 exhibited higher resistance compared to those challenged to Brain Heart Infusion adjusted to the same pH value. Similarly, Waterman and Small

[22] also manifested that *S*. Typhimurium was protected when inoculated into ground beef and boiled egg white but not when rice was used. In general, discrepancies between phenotypes in culture media and food matrices has been found by other authors as well [56].

## Conclusions

In conclusion, pre-exposure of *S*. Enteritidis PT4 to organic (acetic acid) or inorganic (HCl) mild treatments may stimulate acid resistance mechanisms against subsequent extreme acid stress. Nevertheless, this effect cannot be directly extrapolated to acid foods, where other convoluted factors compromise the enhanced acid resistance phenotype. More specifically, the composition of adaptation medium (concentration of UAA, pH) and factors prevailing on the subsequent acid challenge (pH, temperature and other intrinsic but unspecified factors), may collectively determine the acid adaptive response of *Salmonella* in foods and, thus, alter the resistance phenotypes. Further work is required in order to elucidate the effect of the food compounds in the total acquired acid resistance. In addition, given that strain variations can dramatically affect the acid resistance phenotypes, experimental assays expanded to include more strains adapted under the condition employed in the present study should be prompted.

## Supporting information

**S1 Table. Inactivation of NA and adapted *S*. Enteritidis PT4 cells at TSB$_{2.5}$.** Different letters within columns at a given pH value indicate statistical differences among treatments at the same time intervals according to Tukey's HSD test.
(PDF)

**S2 Table. Exposure to TSB$_{4.35}$ at 37°C adjusted with citric acid.** Different letters within the same row indicate statistical differences according to Tukey's HSD test.
(PDF)

**S3 Table. Exposure to TSB$_{4.35}$ at 37°C adjusted with HCl.** Different letters within the same row indicate statistical differences according to Tukey's HSD test.
(PDF)

**S4 Table. Exposure to TSB$_{4.35}$ at 5°C adjusted with citric acid.** Different letters within the same row indicate statistical differences according to Tukey's HSD test.
(PDF)

**S5 Table. Exposure to TSB$_{4.35}$ at 5°C adjusted with HCl.** Different letters within the same row indicate statistical differences according to Tukey's HSD test.
(PDF)

**S6 Table. Exposure to tarama salad stored at 5°C.** Different letters within the same row indicate statistical differences according to Tukey's HSD test.
(PDF)

**S1 Fig. pH changes during storage of adapted and NA cells of *S*. Enteritidis in tarama salad stored at 5°C.**
(PDF)

## Acknowledgments

The authors would like to thank Laboratory of Food Microbiology and Biotechnology, Agricultural University of Athens, Greece, for the kindly provision of *Salmonella* Enteritidis Phage Type 4 P167807 strain.

## Author Contributions

**Conceptualization:** Alkmini Gavriil, Athina Thanasoulia, Panagiotis N. Skandamis.

**Data curation:** Alkmini Gavriil, Panagiotis N. Skandamis.

**Formal analysis:** Alkmini Gavriil, Panagiotis N. Skandamis.

**Investigation:** Alkmini Gavriil, Athina Thanasoulia.

**Methodology:** Alkmini Gavriil, Athina Thanasoulia, Panagiotis N. Skandamis.

**Resources:** Panagiotis N. Skandamis.

**Writing – original draft:** Alkmini Gavriil, Panagiotis N. Skandamis.

**Writing – review & editing:** Alkmini Gavriil, Athina Thanasoulia, Panagiotis N. Skandamis.

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
