## [Decision Letter · Decision Letter 0]

7 May 2020

PONE-D-20-07602

Studying the effect of undissociated acetic acid and pH adaptation on acid resistance of Salmonella enterica sub. enterica serovar Enteritidis Phage Type 4: implications of food matrix associated factors

PLOS ONE

Dear phD candidate Gavriil,

Thank you for submitting your manuscript to PLOS ONE. After careful consideration, we feel that it has merit but does not fully meet PLOS ONE’s publication criteria as it currently stands. Therefore, we invite you to submit a revised version of the manuscript that addresses the points raised during the review process.

We would appreciate receiving your revised manuscript by Jun 21 2020 11:59PM. To enhance the reproducibility of your results, we recommend that if applicable you deposit your laboratory protocols in protocols.io, where a protocol can be assigned its own identifier (DOI) such that it can be cited independently in the future. For instructions see: http://journals.plos.org/plosone/s/submission-guidelines#loc-laboratory-protocols

We look forward to receiving your revised manuscript.

Kind regards,

Leonidas Matsakas

Academic Editor

PLOS ONE

Journal Requirements:

2. In your Methods section, please provide additional details regarding each of the cell lines used in your study, including any quality control testing procedures, and ensure you have described the source. For more information regarding PLOS' policy on materials sharing and reporting, see https://journals.plos.org/plosone/s/materials-and-software-sharing#loc-sharing-materials, and for more information on PLOS ONE's guidelines for research using cell lines, see https://journals.plos.org/plosone/s/submission-guidelines#loc-cell-lines. Also please provide additional details regarding the food materials used in your study and ensure you have described the source.

Reviewers' comments:

Reviewer's Responses to Questions

**Comments to the Author**

1. Is the manuscript technically sound, and do the data support the conclusions?

Reviewer #1: Yes

Reviewer #2: Partly

2. Has the statistical analysis been performed appropriately and rigorously? 

Reviewer #1: Yes

Reviewer #2: Yes

3. Have the authors made all data underlying the findings in their manuscript fully available?

Reviewer #1: Yes

Reviewer #2: Yes

4. Is the manuscript presented in an intelligible fashion and written in standard English?

Reviewer #1: Yes

Reviewer #2: Yes

5. Review Comments to the Author

Reviewer #1: Line 54. Missing parenthesis;

Line 108-109. Indicate the country of production of the acetic acid and sodium hydroxide;

Line 113. Should be deleted “or no growth”;

Line 185. Where is the detection limit known? Literature source?;

Determine the inherit micro flora paragraph (Line 190-193) could be integrated in paragraph (173-186) or specified somewhere separately;

Line 306-307. Table 1. It should be noted that acetic acid buffers may be used?

At least half of the references should be updated from 2015-2020;

Reviewer #2: This manuscript reports the effect of undissociated acetic acid on the induction of cell acid resistance in Salmonella sv Enteritidis. Authors prepared adaptation media by adding acetic acid to TSB to different concentrations, and further adjusting the media pH to designated levels, by which they calculated the concentration of undissociated acetic acid for each acetate/pH combination. A S. Enteritidis strain grown in these adaptation media was challenged at pH 2.5 and the survival kinetics was used to assess the effect of adaptation on acid resistance. Authors also used a Greek acidic food, Tarama, to evaluated the effect of the adaptation on acid resistance. In addition, different acidulants and temperature were examined for the survival of acid adapted Salmonella.

The research objectives and the approaches seem reasonable. The outcomes are mixed. The adaptation in some acetate/pH combinations (hence different undissociated acetic acid concentrations) seemed to enhance acid resistance, extending the time of 4 log reduction by few minutes when challenged at pH2.5. But this reviewer failed to see a clear trend for this effect. The adaptation did not affect the survival of S. Enteritidis in Tarama.

The presentation needs lot more polishing before ready for submission. Overall (especially so for the methods and results sections), it reads like a graduate student’s earlier draft without too much input from the professor(s). Besides the numerous grammar and wording errors, the manuscript is somewhat disconnected among different sections. Some parts, especially the methods and results sections, need to be better delineated.

The following are editorial comments:

L1. Delete “Studying for”. Overall, the title does not seem to capitulate the essence of the manuscript. Besides pH, what other food matrix associated factors were examined?

L60. What does “composition” mean here?

L70. “the gastrointestinal human or animal track”?

L71-72. “Despite their antibacterial activity, pathogens have…”. What does “their” refer to? I believe authors mean organic acids, but in this sentence, it would mean pathogens.

L79. “it is obvious that consumer safety renders questionable.” Very unclear. What does consumer safety render questionable? Or what does render consumer safety questionable?

L82. What is the “drastic form” of the acid?

L86. “This serotype was selected as the most prevalent in Europe”. Change “as” to “as it was”, or “because it was”, etc.

L88. It does not seem “transferability” is the right word.

L94. There is only a single strain. Also, change the order to Bacteria strain and growth conditions.

L97-98, 100-101, etc. What does LabM etc mean? If that was the provider information, use the common format, as name, city, country). State incubation when giving temperatures and time information. Lab book information is not needed in the manuscript.

L104 etc. Capitalizer “ringer”.

L117-144. This section talks about several things. May need additional subheadings to better delineate the procedures.

L125. The use of “in vitro” or “in situ” etc is confusing and technically wrong. Suggest to avoid.

L170, 179. Should “4” be 40?

L183. What was done to make sure that those enumerated were S. enteritidis colonies on TSA plate, especially considering that the same plate was used for enumerating native microbiota (L191)?

L188, 190. Change “inherent” to indigenous, as in discussion.

L220-222. If MIC was 5.2-7.2, and media with undissociated acid>7.2 mM were bactericidal, how was the adaptation in these media achieved? What was the initial inoculation in the adaptation media?

L290-291. “(6.9 mM, treatment 45mM/pH6.0)” Is this correct?

L312-314. Does this mean some of the values are from one model, while others from another model?

L317. Change “independently” to regardless?

L366. “at 4mM/pH6.0”. Is this correct?

6. PLOS authors have the option to publish the peer review history of their article (what does this mean?). If published, this will include your full peer review and any attached files.

Reviewer #1: No

Reviewer #2: No

---

## [Author Response · Author response to Decision Letter 0]

1 Jun 2020

Reviewer 1: We have amended the manuscript taking your suggestions into consideration. We would like to thank you for your valuable comments that considerably improved the presentation of our work. 

Reviewer 2: We have amended the manuscript taking your suggestions into consideration. We have carefully and extensively revised the manuscript and we would like to thank you for your constructive comments that helped us greatly improve the presentation of our work.

---

## [Editor Report · Decision Letter 1]

8 Jun 2020

Sublethal concentrations of undissociated acetic acid may not always stimulate acid resistance in Salmonella enterica sub. enterica serovar Enteritidis Phage Type 4: implications of challenge substrate associated factors

PONE-D-20-07602R1

Dear Dr. Skandamis,

We’re pleased to inform you that your manuscript has been judged scientifically suitable for publication and will be formally accepted for publication once it meets all outstanding technical requirements.

Kind regards,

Leonidas Matsakas

Academic Editor

PLOS ONE
---

## [Editor Report · Acceptance letter]

10 Jul 2020

PONE-D-20-07602R1 

Sublethal concentrations of undissociated acetic acid may not always stimulate acid resistance in *Salmonella enterica* sub. *enterica* serovar Enteritidis Phage Type 4: implications of challenge substrate associated factors 

Dear Dr. Skandamis:

I'm pleased to inform you that your manuscript has been deemed suitable for publication in PLOS ONE. Congratulations! Your manuscript is now with our production department. 

Kind regards, 

on behalf of

Dr. Leonidas Matsakas 

Academic Editor

PLOS ONE